# Postprandial Blood Pressure Decrease in Patients with Type 2 Diabetes and Mild or Severe Cardiac Autonomic Dysfunction

**DOI:** 10.3390/ijerph16050812

**Published:** 2019-03-06

**Authors:** Masahiko Hashizume, Saori Kinami, Keiichi Sakurai, Kazuhiro P. Izawa, Hideyuki Shiotani

**Affiliations:** 1Graduate School of Health Sciences, Kobe University, Kobe 654-0142, Japan; izawapk@harbor.kobe-u.ac.jp (K.P.I.); hshio@kobe-u.ac.jp (H.S.); 2Department of Rehabilitation, Arima Onsen Hospital, Kobe 651-1401, Japan; 3Department of General Internal Medicine, Akashi Medical Center, Akashi, Hyogo 674-0063, Japan; redscorpio102@gmail.com; 4Department of Internal Medicine, Tatsuno Central Hospital, Tatsuno, Hyogo 679-4121, Japan; ksakurar@hotmail.com

**Keywords:** postprandial hypotension, diabetic autonomic neuropathy, blood pressure, type 2 diabetes mellitus

## Abstract

*Background* Few reports have evaluated the relationship between changes in postprandial blood pressure and the severity of autonomic dysfunction in patients with type 2 diabetes. This was a cross-sectional study designed to investigate postprandial blood pressure changes in individuals without type 2 diabetes and patients with type 2 diabetes and mild or severe cardiac autonomic dysfunction. *Methods* Forty patients with type 2 diabetes mellitus and 20 individuals without type 2 diabetes participated in this study. Fifty-two participants underwent a meal tolerance test. Blood pressure (brachial systolic blood pressure (bSBP) and central systolic blood pressure (cSBP)), electrocardiogram recordings, and blood samples were assessed before and after meal ingestion. Patients with diabetes were divided into two groups based on their coefficient of variation of R–R intervals (CVRR): a normal or mildly dysfunctional group (mild group, CVRR ≥ 2%; *n* = 20) and a severely dysfunctional group (severe group, CVRR < 2%; *n* = 15). *Results* In the control group, bSBP and cSBP did not significantly change after meal ingestion, whereas both decreased significantly at 60 min after meal ingestion in the mild and severe groups. While blood pressure recovered at 120 min after meal ingestion in the mild group, a significant decrease in blood pressure persisted at 120 min after meal ingestion in the severe group. *Conclusions* Based on these results, adequate clinical attention should be paid to the risk of serious events related to postprandial decreases in blood pressure, particularly in patients with diabetes and severe cardiac autonomic dysfunction.

## 1. Introduction

Diabetes mellitus (DM) is a chronic disease characterized by hyperglycemia resulting from defects in insulin secretion, action, or both [1]. Long-term complications of DM include retinopathy, nephropathy, and neuropathy [1]. Diabetes mellitus is a growing global concern, with the International Diabetes Foundation predicting that the number of people with diabetes will increase to 629 million worldwide by 2045 [2].

Previous studies have shown that the prevalence of hypertension is increased in patients with DM [3]. Tight control of blood pressure (maintenance below 130/80 mmHg) is clinically recommended in patients with type 2 diabetes mellitus (T2DM) and hypertension by the Japanese Society of Hypertension’s 2014 guidelines [4]. Therefore, many patients with DM and hypertension develop hypotension as an undesired effect of blood pressure control measures [5].

A critical symptom of hypotension in patients with DM is postprandial hypotension (PPH) [6]. Postprandial hypotension is commonly defined as a decrease of more than 20 mmHg in systolic blood pressure (SBP) within 2 h of the start of a meal, and it is an important clinical problem that disposes patients to syncope, falls, angina pectoris, and cerebrovascular events [7,8]. Tabara et al. [9] previously reported that lacunar infarctions were significantly more common in elderly individuals with a postprandial decrease of more than 10 mmHg in systolic blood pressure (SBP) than was found in controls. Sasaki et al. [6] also suggested that PPH might be an important clinical symptom of cardiovascular disturbances in patients with DM. In general, PPH may be more common and further associated with more severe outcomes in patients with DM and autonomic dysfunction [10]. Although the precise pathophysiological mechanisms underlying PPH remain unknown, Tanakaya et al. [11] have suggested that a lack of compensatory sympathetic activation contributes to PPH in patients with DM. Given this, PPH may be related to autonomic dysfunction. However, only a few reports have evaluated the clinical relationship between changes in postprandial blood pressure and the severity of autonomic dysfunction.

Further, for blood pressure measurements, the importance of measuring central systolic blood pressure (cSBP), as performed in this study, is that this measure is more closely related to cardiovascular events compared to peripheral blood pressure and is recognized by studies such as the Anglo-Scandinavian Cardiac Outcomes Trial Conduit Artery Function Endpoint (ASCOT-CAFE) study [12,13]. However, studies of PPH that include cSBP as an outcome have not been performed yet.

Given this gap in the existing research, we hypothesized that there might be a difference in the change in postprandial blood pressure in patients with T2DM and mild or severe cardiac autonomic dysfunction when measuring both brachial systolic blood pressure (bSBP) and cSBP and that there would be a positive correlation between the decrease in postprandial blood pressure and the severity of autonomic dysfunction.

The purpose of the present study was as follows: (1) to investigate differences in the change in postprandial blood pressure in individuals without T2DM and patients with T2DM and mild or severe cardiac autonomic dysfunction by measuring brachial systolic blood pressure (bSBP) as well as cSBP; and (2) to clarify the relationship between changes in postprandial blood pressure and the severity of autonomic dysfunction.

## 2. Materials and Methods

### 2.1. Study Design and Participants

This was a cross-sectional study, which included 40 patients with T2DM and 20 individuals without T2DM. Patients with T2DM were recruited from outpatients and inpatients at Akashi Medical Center (Akashi, Hyogo, Japan) and Tatsuno Central Hospital (Tatsuno, Hyogo, Japan). All patients were diagnosed according to the Japan Diabetes Society’s T2DM criteria [14]. Study inclusion criteria included a patient age of 40–80 years and 6.0% ≤ HbA1c ≤ 12.0%. Study exclusion criteria included a diagnosis of type 1 diabetes, comorbid cardiovascular or other vascular diseases, arrhythmias such as atrial fibrillation, severe liver dysfunction, severe renal disease (estimated glomerular filtration rate < 30 mL/min/1.73 m^2^), pregnancy, and treatment with a β-blocker. We also recruited 20 individuals without T2DM through advertisements at the Graduate School of Health Sciences, Kobe University (Kobe, Japan). The only control group inclusion criterion was an age of 40–80 years. Control group exclusion criteria included pre-existing diabetes, cardiovascular disease, arrhythmias such as atrial fibrillation, pregnancy, being a night-shift worker, and smoking. The study was approved by the Ethics Committee of the Graduate School of Health Sciences, Kobe University (approval no. 14). The purpose and risk of this study was explained to each participant before their written informed consent was obtained.

### 2.2. Study Protocol

Figure 1 depicts the experimental protocol and timeline. Participants were asked to refrain from consuming food and drink that contained caffeine or alcohol and were instructed not to perform any exercise after 22:00 the previous night. They were asked to finish breakfast by 7:30 on the day of measurement. The administration of any hypertensive drugs was prohibited at least 24 h before the study. Patients were also advised not to take any antidiabetic drugs (including insulin) from the evening before the study until the study was completed. Before the assessment, height and body weight were measured. We measured the height with the participants’ shoes off, and the weight with light clothing and empty pockets.

Assessments began at 11:30. After 10 min of rest in a seated position, blood pressure and radial arterial waveforms were measured. The measurement of cSBP was performed simultaneously. An electrocardiogram (ECG) was then obtained while the patient was in a supine position. After this, intravenous blood samples were collected from the brachial vein. Starting at 12:00, participants were served a test meal (E460F18, Kewpie, Tokyo, Japan; 460 kcal, 56.5 g of carbohydrate, 18.0 g of protein, 18.0 g of fat, and 670 mg of sodium). The participants were those who had completely consumed a meal within 15–20 min. The same set of measurements was then repeated at 60 (13:00) and 120 (14:00) minutes after the participants began their meal. During the measurement period, participants were allowed to drink only water (less than 500 mL) in addition to the test meal.

### 2.3. Data Collection

Brachial systolic blood pressure, brachial diastolic blood pressure (bDBP), and cSBP were measured using a digital automated sphygmomanometer (HEM-9000AI, Omron Healthcare, Kyoto, Japan). Brachial systolic blood pressure and bDBP measurements were recorded using a cuff around the right upper arm. Pressure waveforms in the radial artery were recorded non-invasively using a tonometric approach. Early systolic pressure (SBP1), late systolic pressure (SBP2), and augmentation index (AI) were also measured and calculated automatically by the HEM-9000AI using the following equations: AI = (SBP2 − DBP)/(SBP1 − DBP), SBP2 (mmHg) = AI × (SBP − DBP) + DBP. Because Takazawa et al. [15] demonstrated a good correlation between cSBP and SBP2, cSBP calculations were based on the SBP2 values obtained in the present study. Furthermore, bSBP, bDBP, and cSBP were each measured twice and the average values were used for further analyses. Electrocardiogram recordings (Cardio Star FX-7432, Fukuda Denshi, Tokyo, Japan) were obtained across 200 pulses using lead II. R–R intervals of 200 pulses recorded on the ECG, coefficient of variation of R–R intervals (CVRR), and average heart rates (HRs) were calculated as follows: CVRR = (standard deviation/mean value of R–R intervals) × 100 (%). These assessments were performed by a medical technologist and a trained assistant. Venous blood samples (10 mL) were obtained by a medical doctor. Plasma glucose, serum insulin, triglycerides, high-density lipoprotein (HDL) cholesterol, and low-density lipoprotein (LDL) cholesterol levels were measured by standard methods using an autoanalyzer.

### 2.4. Statistical Analyses

Data were expressed as the mean value ± standard error. Body mass index (BMI) was calculated using the following equation: BMI = body weight (kg)/height (m)^2^. We divided the patients based on their CVRR into two groups: a normal or mild cardiac autonomic dysfunction (mild group; CVRR ≥ 2%) and a severe cardiac autonomic dysfunction (severe group; CVRR < 2%). A CVRR value of 2% is considered the critical level below which diabetic autonomic neuropathy occurs [16]. Differences among groups at baseline were evaluated using a one-way analysis of variance (ANOVA) and the Bonferroni post-hoc test. Differences between the mild and severe groups at baseline were evaluated using *t*-tests and Fisher’s exact test. Changes in measured variables following meal ingestion were examined using two-way ANOVAs for repeated measures and the Bonferroni post-hoc test. Pearson’s correlation analyses were performed to identify correlations between preprandial blood pressure (bSBP and cSBP), and the magnitude of postprandial change in bSBP over 1 or 2 h (∆bSBP1hr, or ∆bSBP2hrs) or cSBP over 1 or 2 h (∆cSBP1hr, ∆cSBP2hrs), calculated by subtracting preprandial blood pressure from postprandial blood pressure. A correlation analysis was also used to assess any correlation between preprandial CVRR and the magnitude of postprandial blood pressure change (∆bSBP1hr, ∆bSBP2hrs, ∆cSBP1hr, and ∆cSBP2hrs).

A stepwise multiple linear regression analysis was performed to evaluate the independent variables that were associated with the magnitude of postprandial change in blood pressure for all subjects. The independent variables were ∆bSBP1hr, ∆bSBP2hrs, ∆cSBP1hr, and ∆cSBP2hrs. The following factors were used as dependent variables: age, BMI, CVRR, bSBP, and plasma glucose at baseline. The effectiveness of the models was assessed by *p*-values, and the completeness of the models were assessed by the coefficients of determination. Statistical analyses were conducted using SPSS version 20 (IBM, Chicago, IL, USA). A *p*-value of less than 0.05 was considered to indicate statistical significance.

## 3. Results

Figure 2 shows the participant flow through this study, which included 35 patients and 17 individuals. The mild group (CVRR ≥ 2%) was composed of 20 patients, and the severe group (CVRR < 2%) was composed of 15 patients. Baseline patient characteristics are shown in Table 1 and Table 2. In this study, 21 patients were treated with diet and exercise only, 14 patients were administered oral hypoglycemic agents, and eight patients received insulin therapy. In addition, 14 patients received the following drugs for treatment of hypertension: angiotensin-receptor blockers (*n* = 12) and calcium channel blockers (*n* = 8). The mean CVRR in the severe group was 1.45 ± 0.11%, which was significantly lower than that in the mild group (3.23% ± 0.19%, *p* < 0.01). In addition, the mean CVRR in the mild group was significantly lower than that in the control group (*p* < 0.05). There were no significant differences in age or measures of blood pressure, such as bSBP and cSBP, among the groups.

During meal tolerance testing, plasma glucose significantly increased at 60 min in all groups (Figure 3A). Serum insulin concentrations also significantly increased at 60 min in the control and mild groups, while no significant difference was observed in the severe group (Figure 3B).

After meal ingestion in the control group, bSBP and cSBP did not significantly change, while HR significantly increased at 60 and 120 min (Figure 4). In the mild and severe groups, bSBP significantly decreased at 60 min (from 136.5 ± 3.35 to 127.8 ± 2.78 mmHg in the mild group, *p* < 0.01; from 131.8 ± 3.87 to 122.6 ± 3.22 mmHg in the severe group, *p* < 0.01). This decrease in bSBP recovered by 120 min in the mild group (134.3 ± 3.04 mmHg), whereas a significant decrease in bSBP persisted at 120 min in the severe group (122.2 ± 3.51 mmHg, *p* < 0.01, Figure 4A). Similarly, cSBP significantly decreased in the mild and severe groups at 60 min after meal ingestion (from 141.5 ± 3.56 to 130.7 ± 2.99 mmHg in the mild group, *p* < 0.01; from 135.9 ± 4.11 to 122.8 ± 3.46 mmHg in the severe group, *p* < 0.01); the decrease in cSBP recovered at 120 min in the mild group (138.1 ± 3.22 mmHg), whereas a significant decrease in cSBP persisted at 120 min in the severe group (124.6 ± 3.72 mmHg, *p* < 0.01, Figure 4B). HR did not significantly change in either the mild or severe groups (Figure 4C).

Pearson’s correlation analysis revealed that ∆bSBP1hr was significantly and negatively correlated with bSBP at baseline (*r* = −0.519, *p* < 0.001), whereas ∆bSBP2hrs was not. Similarly, ∆cSBP1hr was significantly and negatively correlated with cSBP at baseline (*r* = −0.377, *p* = 0.006), whereas ∆cSBP2hrs was not. ∆bSBP2hrs (*r* = 0.415, *p* = 0.003) and ∆cSBP2hrs (*r* = 0.430, *p* = 0.002) were significantly and positively correlated with CVRR, whereas ∆bSBP1hr and ∆cSBP1hr did not significantly correlate with CVRR.

As shown in Table 3, multiple linear regression analyses revealed that bSBP at baseline was a significant predictor of ∆bSBP1hr and had a greater influence than CVRR, as indicated by a higher standardized coefficient. In addition, ∆bSBP2hrs was independently related to CVRR, whose standardized coefficient also revealed a greater influence than bSBP. Similarly, ∆cSBP1hr was independently related to cSBP at baseline, and ∆cSBP2hrs was independently related to CVRR with a higher standardized coefficient that indicated a greater influence than cSBP.

## 4. Discussion

The main findings of this study are as follows: (1) cSBP, as well as bSBP, decreased after meal ingestion in patients with T2DM and both mild and severe cardiac autonomic dysfunction; furthermore, the decline in bSBP and cSBP at 60 min after meal ingestion correlated with preprandial bSBP and cSBP, respectively. (2) Postprandial decreases in both bSBP and cSBP persisted longer in patients with T2DM and severe cardiac autonomic dysfunction, resulting in a magnitude of decline in bSBP and cSBP at 120 min after meal ingestion that was independently associated with CVRR. These results suggest that adequate clinical attention should be paid to possible serious events related to PPH, especially in patients with T2DM and severe cardiac autonomic dysfunction.

In this study, bSBP decreased at 60 min after meal ingestion not only in patients with severe autonomic dysfunction but also in those with mild autonomic dysfunction, suggesting that a postprandial decrease in blood pressure might occur in the early stage of T2DM. In the present study, blood pressure was maintained and HR was increased in middle-aged individuals without diabetes after meal ingestion. The difference in postprandial decreases in blood pressure between individuals without T2DM and patients with T2DM could be attributed to differences in their compensatory response to postprandial hemodynamic change [7]. That is, in the individuals without diabetes, HR increased significantly after a meal to compensate for the decreased blood pressure, which resulted to a stable postprandial blood pressure. On the contrary, such an increase in HR was not observed in patients with T2DM. This lack of HR increase might lower the blood pressure.

Our study results are in concordance with those of previous studies. For instance, some studies have found that postprandial blood pressure was maintained in healthy participants despite vasodilation and splanchnic blood pooling caused by gastrointestinal vasoactive peptides [17,18,19]. In healthy participants, HR and cardiac output increased significantly after meal ingestion to compensate for the lowering of blood pressure, which resulted in the maintenance of postprandial blood pressure [7,17]. However, in patients with T2DM, these compensatory responses may be diminished by cardiac autonomic dysfunction such that blood pressure cannot be maintained. Indeed, in the present study, CVRR was lower in the mild and severe groups than in the control group, revealing that cardiac autonomic dysfunction may occur in T2DM. In addition, there were no significant changes in HR in patients with T2DM and either mild or severe cardiac autonomic dysfunction. Previous studies have similarly reported that meal ingestion resulted in blood pressure reductions without changes in cardiac output or HR in patients with DM [11,20,21]. Some studies have also suggested that several sympathetic cardiovascular indexes increased in response to meal ingestion in healthy participants, whereas these compensatory responses were diminished in patients with autonomic dysfunction [17,22]. These studies are consistent with our findings, collectively suggesting that postprandial decreases in blood pressure in patients with T2DM may be due to cardiac autonomic dysfunction.

In the present study, we found that ∆bSBP1hr significantly correlated with preprandial bSBP. Previous studies have made similar observations among elderly or hypertensive patients. For example, Vaitlevicius et al. [23] reported that maximal postprandial reductions in SBP occurred between 45 and 60 min and were inversely correlated with preprandial SBP values in elderly nursing-home residents. Puisieux et al. [24] also reported this relationship between preprandial SBP and postprandial blood pressure decline and suggested that higher preprandial SBP levels were associated with greater postprandial blood pressure declines and an elevated risk of PPH in elderly individuals. Mitro et al. [18] further observed a high prevalence of PPH at 60 min in hypertensive patients and suggested that the decrease in baroreflex sensitivity associated with blood pressure elevations might lead to PPH. Collectively, the conclusion that PPH is associated with hypertension may support our findings here. Furthermore, the present study suggests that postprandial decreases in SBP at 60 min in patients with T2DM may be related to hypertension rather than the degree of cardiac autonomic dysfunction.

Although the precise mechanisms which result in PPH have not been clearly elucidated, earlier studies [19,25,26,27,28,29,30,31] had revealed several hemodynamic changes after meal ingestion. Meal ingestion promotes the secretion of vasodilators like insulin and neurotensin [8,19,25,26,27,28,29,30,31]. These agents cause vasodilation and splanchnic blood pooling, which can lead to a reduction in cardiac output, and thus, may contribute to the development of PPH [7,25]. In the present study, serum insulin increased after meal ingestion in patients with mild cardiac autonomic dysfunction but did not significantly change in patients with severe dysfunction. Despite these differences in insulin levels after meal ingestion, decreases in blood pressure were similar in patients with both mild and severe dysfunction at 60 min. Some previous studies have showed that exogenous insulin administration may lower blood pressure and sometimes result in syncope in patients with T2DM [26,27]. Insulin has further been implicated in the etiology of postprandial hypotension [28]. However, the results reported here suggest that insulin may not play a major role in PPH in patients with T2DM. Alternatively, eating possibly promotes neurotensin secretion from the small intestine in healthy participants and patients with autonomic failure [29,30]. In addition, voglibose, an alpha-glucosidase inhibitor, prevented PPH by reduced splanchnic blood pooling due to an inhibition of neurotensin release in patients with neurologic disorders [31]. Therefore, neurotensin, rather than insulin, might be the primary driver of PPH pathogenesis [19].

We also found that CVRR was independently related to ∆bSBP2hr and had a greater influence than bSBP. This suggests that more severe cardiac autonomic dysfunction might delay recovery from decreases in blood pressure. To the best of our knowledge, this study is the first to report that postprandial decrease in blood pressure persists longer in patients with T2DM and severe cardiac autonomic dysfunction. Furthermore, impaired sympathetic activation due to severe cardiac autonomic dysfunction might not increase peripheral resistance, which was diminished by vasodilation after meal ingestion. Masuda et al. [32] suggested that to prevent PPH, sympathetic nervous activity during eating needs to be two to three times higher than that which occurs, on average, during daily activity. Therefore, in patients with T2DM and severe cardiac autonomic dysfunction, sympathetic dysfunction after meal ingestion might persist longer, resulting in PPH. Moreover, vasodilator secretion may persist longer due to delayed gastric emptying in these patients and may thus account for persistent peripheral vasodilation and decreases in blood pressure in patients with severe dysfunction. However, further studies are needed to investigate differences in the compensatory mechanisms of patients with T2DM and mild autonomic complications compared to those with severe autonomic complications.

In the present study, as with bSBP, cSBP also decreased in patients with both mild and severe autonomic dysfunction. The decrease in the severe group persisted for up to 120 min after meal ingestion. Therefore, we extended the findings observed in bSBP to cSBP. As mentioned above, in general, cSBP is recognized to be more closely related to cardiovascular events compared to peripheral blood pressure, as demonstrated by studies such as ASCOT-CAFE study [12,13]. cSBP is considered to be the blood pressure at the aortic root, which is directly applied to key organs, such as the heart and brain. Previous reports have described how elevation of the cSBP induces coronary arteriosclerosis [33,34]. As the observation and reduction of cSBP contribute to the prevention of cardiovascular events, measurement of not only brachial blood pressure but also cSBP may be useful in the prevention and treatment of cardiovascular diseases [15]. This finding of the present study is of clinical significance because it suggests that the persistence of postprandial decrements to cSBP may not only result in dizziness and lightheadedness but may also trigger transient ischemic attacks [35]. Our results revealed an obvious persistence in postprandial decreases in cSBP in patients with T2DM and severe autonomic dysfunction, suggesting that the influence of postprandial decrease in blood pressure may be even greater than might be expected. Given these results, adequate attention should be paid to PPH, especially in patients with T2DM and severe autonomic dysfunction.

Our study has some limitations that warrant discussion. First, we did not measure gastrointestinal hormones beyond insulin. Our results suggest that insulin might not play a major role in PPH in patients with T2DM. It is necessary to further investigate other hormonal mechanisms and the rate of gastric emptying and mesenteric blood flow to clarify additional potential mechanism of PPH. Second, as blood pressure was only measured for 120 min after meal ingestion, blood pressure recovery from hypotension was not observed in patients with severe autonomic dysfunction. While postprandial decreases in blood pressure recovered at 120 min after meal ingestion in patients with mild autonomic dysfunction, this decrease in blood pressure persisted in patients with severe autonomic dysfunction. Had we taken measurements for a longer period of time after meal ingestion, we might have been able to clarify the point at which postprandial hypotension recovers in patients with severe autonomic dysfunction. Third, the present study’s sample size was relatively small. In addition, we did not perform power calculations to determine the number of subjects. Fourth, in the control group, there was no established definition of CVRR, and an oral glucose tolerance test (OGTT) or HbA1c evaluation were not performed. In the control participants, the plasma glucose before meal ingestion was 99.3 ± 1.81 mg/dL, and none had levels exceeding 110 mg/dL, which is the criterion for impaired fasting glucose (IFG). Because we did not perform an OGTT in the control group, we cannot exclude the possibility that individuals with impaired glucose tolerance (IGT) were included. While this is a limitation, the plasma glucose at 120 min after meal ingestion was 108.5 ± 6.59 mg/dL, and no individuals had levels higher than 140 mg/dL. Therefore, we believe that it is unlikely that individuals with IGT were included in the control group and that the probability of misclassification bias is low. Finally, while numerous mechanisms may account for blood pressure reduction after meal ingestion, these remain poorly understood and worthy of additional investigation.

## 5. Conclusions

The present study revealed that postprandial decreases in bSBP and also cSBP occurred in patients with T2DM and both mild and severe cardiac autonomic dysfunction. Furthermore, we found that postprandial decreases in blood pressure persisted for longer in patients with T2DM and severe cardiac autonomic dysfunction. In addition, cardiac autonomic dysfunction was associated with the magnitude of blood pressure decline at 120 min after meal ingestion. Given these findings, attention should be paid to the possible serious events related to PPH, especially in patients with T2DM and severe cardiac autonomic dysfunction.

## Figures and Tables

**Figure 1 ijerph-16-00812-f001:**
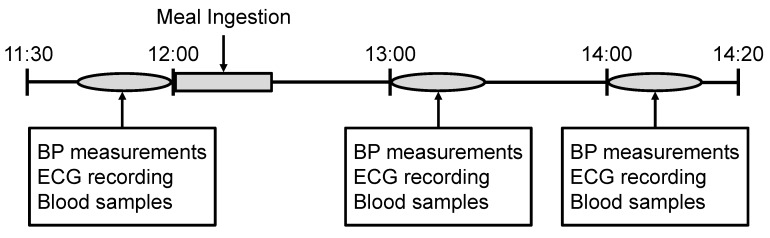
Experimental protocol. Abbreviations: BP, blood pressure; ECG, electrocardiogram.

**Figure 2 ijerph-16-00812-f002:**
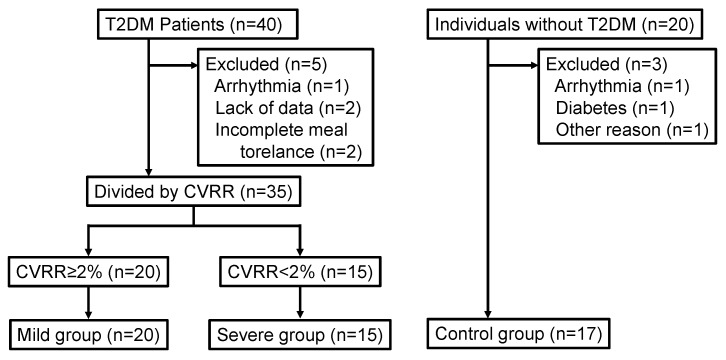
Participant flow. Abbreviations: T2DM, type 2 diabetes mellitus; CVRR, coefficient of variation of R-R intervals.

**Figure 3 ijerph-16-00812-f003:**
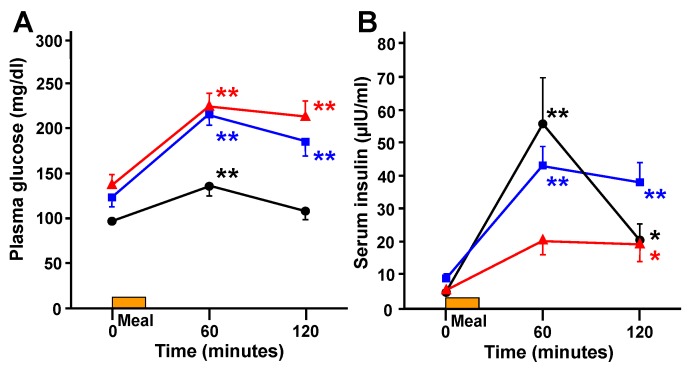
Changes in plasma glucose (**A**) and serum insulin (**B**) during the meal tolerance test. The control group is indicated by the black line and circles (●); the mild group by the blue line and squares (■); and the severe group by the red line and triangles (▲). ** *p* < 0.01, * *p* < 0.05 vs. 0 min.

**Figure 4 ijerph-16-00812-f004:**
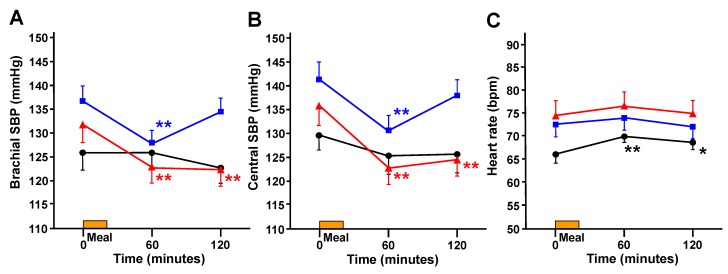
Changes in brachial systolic blood pressure (SBP) (**A**), central SBP (**B**), and heart rate (**C**) during the meal tolerance test. The control group is indicated by the black line and circles (●); the mild group by the blue line and squares (■); and the severe group by the red line and triangles (▲). ** *p* < 0.01, * *p* < 0.05 vs. 0 min.

**Table 1 ijerph-16-00812-t001:** Baseline characteristics.

	Control (*n* = 17)	Mild Group (*n* = 20)	Severe Group (*n* = 15)	*p*-Value
Sex (male/female)	13/4	9/11	6/9	
Age (years old)	54.8 ± 1.58	60.6 ± 1.57	60.1 ± 2.17	0.05
Body mass index (kg/m^2^)	22.0 ± 0.47	25.1 ± 0.77 *	25.8 ± 1.01 **	0.002
bSBP (mmHg)	125.9 ± 3.53	136.5 ± 3.43	131.8 ± 3.75	0.11
bDBP (mmHg)	81.9 ± 2.41	83.5 ± 2.07	81.1 ± 2.31	0.76
cSBP (mmHg)	129.8 ± 3.16	141.5 ± 3.86	135.9 ± 3.60	0.72
HR (bpm)	66.1 ± 1.83	72.6 ± 2.69	74.4 ± 3.52	0.09
CVRR (%)	4.13 ± 0.30	3.23 ± 0.19 *	1.45 ± 0.11 ** ^††^	<0.001
Plasma glucose (mg/dL)	99.3 ± 1.81	124.4 ± 8.52	139.0 ± 12.8 *	0.03
Serum insulin (µIU/mL)	5.53 ± 0.95	9.16 ± 1.50	5.78 ± 1.11	0.09
LDL cholesterol (mg/dL)	125.2 ± 6.92	127.5 ± 7.54	112.2 ± 10.0	0.38
Triglycerides (mg/dL)	98.6 ± 9.68	131.6 ± 17.0	151.0 ± 14.0	0.08
HDL cholesterol (mg/dL)	67.4 ± 3.56	53.4 ± 3.73 *	50.6 ± 3.01 **	0.008

Data given as mean ± SE. ** *p* < 0.01, * *p* < 0.05 vs. Control. ^††^
*p* < 0.01 vs. Mild Group. Abbreviations: bSBP, brachial systolic blood pressure; bDBP, brachial diastolic blood pressure; cSBP, central systolic blood pressure; CVRR, coefficient variation of R–R intervals; HDL, high-density lipoprotein; HR, heart rate; LDL, low-density lipoprotein.

**Table 2 ijerph-16-00812-t002:** Baseline characteristics of patients.

	Mild Group (*n* = 20)	Severe Group (*n* = 15)	*p*-Value
Duration of DM (years)	5.30 ± 2.04	11.0 ± 2.56	0.088
HbA1c (%)	7.05 ± 0.28	8.14 ± 0.52	0.060
Antihypertensive therapy	9 (45%)	5 (33%)	0.521
Angiotensin-receptor blocker	7 (35%)	5 (33%)	0.623
Calcium channel blocker	4 (20%)	4 (27%)	0.428

Data given as mean ± SE. Abbreviations: DM, diabetes mellitus; HbA1c, hemoglobin A1c.

**Table 3 ijerph-16-00812-t003:** Multiple linear regression analysis for predicting the magnitude of postprandial blood pressure change.

Dependent/Independent Variables	B	SE	95%CI	β
∆bSBP1hr				
Model 1: Constant = 38.3; R^2^ = 0.27; adjusted R^2^ = 0.25; *p* < 0.01				
bSBP	−0.33	0.09	−0.51 to −0.16	−0.52
Model 2: Constant = 32.7; R^2^ = 0.34; adjusted R^2^ = 0.31; *p* < 0.01				
bSBP	−0.33	0.08	−0.50 to −0.17	−0.52
CVRR	1.95	0.93	0.08 to 3.83	0.27
∆bSBP2hrs				
Model 1: Constant = −12.2; R^2^ = 0.17; adjusted R^2^ = 0.15; *p* < 0.01				
CVRR	2.82	0.95	0.90 to 4.75	0.42
Model 2: Constant = 14.9; R^2^ = 0.29; adjusted R^2^ = 0.25; *p* < 0.01				
bSBP	−0.20	0.08	−0.36 to −0.05	−0.34
CVRR	2.83	0.90	1.02 to 4.63	0.42
∆cSBP1hr				
Model 1: Constant = 19.8; R^2^ = 0.14; adjusted R^2^ = 0.12; *p* < 0.05				
cSBP	−0.22	0.08	−0.38 to −0.05	−0.38
∆cSBP2hrs				
Model 1: Constant = −13.8; R^2^ = 0.19; adjusted R^2^ = 0.17; *p* < 0.01				
CVRR	2.93	0.95	1.02 to 4.84	0.43
Model 2: Constant = 8.41; R^2^ = 0.26; adjusted R^2^ = 0.23; *p* < 0.01				
cSBP	−0.16	0.08	1.14 to 4.82	−0.28
CVRR	2.98	0.91	−0.32 to −0.01	0.44

Abbreviations: BP, blood pressure; B, unstandardized coefficients; CI, confidence intervals; β, standardized coefficients; bSBP, brachial systolic blood pressure; cSBP, central systolic blood pressure; CVRR, coefficient variation of R–R intervals; SE, standard error; 1 hr, one hour; 2hrs, two hours.

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
