# Peer review of "Postprandial Blood Pressure Decrease in Patients with Type 2 Diabetes and Mild or Severe Cardiac Autonomic Dysfunction"

_ijerph, 2019, doi:10.3390/ijerph16050812_

Round 1

Reviewer 1 Report

This study by Masahiko Hashizume et al to investigated differences in the change in postprandial blood pressure in patients with T2DM and mild or severe cardiac autonomic dysfunction by measuring bSBP as well as cSBP and to clarified the relationship between changes in postprandial blood pressure and the severity of autonomic dysfunction. The present study revealed that postprandial decreases in bSBP and also cSBP occurred in patients with T2DM and both mild and severe cardiac autonomic dysfunction. Furthermore, authors found that postprandial decreases in blood pressure persisted for longer in patients with T2DM and severe cardiac autonomic dysfunction. For the study the presented data are quite sufficient.

Author Response

Response to Reviewer 1 Comments

Thank you very much for your review and for your helpful comments. We have revised this manuscript on the basis of your comments and those of the other reviewers.The revised paper has been edited for grammar and journal style by an expert in English medical writing prior to resubmission. Revisions are highlighted in yellow in the manuscript and are underlined below

Reviewer 2 Report

The authors present an interesting study on the association between type 2 diabetes mellitus (T2DM) and cardiac autonomic dysfunction with postprandial hypotension. The study results showed that T2DM patients with either mild or severe autonomic dysfunction had a significant decrease in brachial and central aortic systolic blood pressure 60 minutes after the given meal, but this decrease was only sustained at 2 hours in T2DM patients with severe autonomic dysfunction. The study was well controlled and the results were clearly presented. However, there are aspects of the study that I think require further clarification:

postprandial hypotension (PPH) is defined as a drop in systolic blood pressure of 20 mmHg or more (ref 7 from the manuscript). This should be stated and referenced in the manuscript

following from the definition of PPH, did the drop in bSBP in the T2DM patients go beyond 20 mmHg? Judging from the graph it seems to be around 20 mmHg, and it would be good if authors can state whether this drop was > 20 mmHg

The definition used for mild autonomic dysfunction in T2DM patients (CVRR > 2%) is very wide and could also be applied to the control group. Although the results do show that this group had a significantly lower CVRR than the control group, is there actually a definition for normal CVRR? In general, do all T2DM patients have lower CVRR than normal? In this particular cohort, did any T2DM patients have a CVRR similar or even higher than individuals in the control group? Were there any T2DM patients in this cohort that actually had normal autonomic function? The distinction is very important and I think these should be clarified, otherwise one cannot conclude that the drop in SBP is entirely due to autonomic dysfunction, but rather it would be a difference between T2DM and control, especially since the absolute magnitude of change in both the mild and severe groups were almost identical from Figure 4.

from the graph (Figure 4) it seems there was a larger decrease in cSBP at 60 minutes compared to bSBP, and that cSBP was recovering slightly more than bSBP at 120 minutes (bSBP almost unchanged)... was cSBP@120 different to cSBP@60? Could the authors discuss these observations and what the implications may be? 

Further to the previous point, the current discussion on cSBP results do not really identify the advantage of measuring cSBP as well as bSBP. The one reference currently used to support the need for measuring cSBP (ref 33) did not actually measure central aortic BP.

although brachial DBP does not pertain to the definition of PPH, it would be interesting to know if it changed after the meal

more discussion on the mechanistic side of how autonomic dysfunction can lead to PPH would be helpful

Other minor comments:

a figure legend may be neater than writing the group names above the relevant lines on the figure

pg 8, line 250, authors stated bSBP at 60 minutes may be due to hypertension rather than degree of autonomic dysfunction. However, CVRR was also independently associated with change in bSBP1hr 

were interactions included in the stepwise regression?

Author Response

Response to Reviewer 2 Comments

Thank you very much for your review and for your helpful comments. We have revised this manuscript on the basis of your comments and those of the other reviewers.The revised paper has been edited for grammar and journal style by an expert in English medical writing prior to resubmission. Revisions are highlighted in yellow in the manuscript and are underlined belowOur responses to your comments are shown below.

Point 1: postprandial hypotension (PPH) is defined as a drop in systolic blood pressure of 20 mmHg or more (ref 7 from the manuscript). This should be stated and referenced in the manuscript

Response 1: As suggested, we have described about the definition of postprandial hypotension in the Introduction section as follows:

P.2, L.49-50. Introduction

PPH is commonly defined as a decrease of more than 20 mmHg in systolic blood pressure (SBP) within 2 hours of the start of a meal, and itis an important clinical problem that disposes patients to syncope, falls, angina pectoris, and cerebrovascular events [7,8].

Point 2:following from the definition of PPH, did the drop in bSBP in the T2DM patients go beyond 20 mmHg? Judging from the graph it seems to be around 20 mmHg, and it would be good if authors can state whether this drop was > 20 mmHg

Response 2: In this study, two patients in the mild group and one patient in the severe group met the definition of postprandial hypotension. In addition, eight patients in the mild group had systolic blood pressure (SBP) decrease of more than 10 mmHg compared to nine patients in the severe group who had SBP decrease of more than 10 mmHg within 2 hours of the start of a meal. Tabara et al. previously reported that lacunar infarctions were significantly more common in elderly individuals with a postprandial decrease of more than 10 mmHg in systolic blood pressure (SBP) than was found in controls [9]. We think that the postprandial BP decrease of more than 10 mmHg might have clinical importance. The magnitude of postprandial blood pressure change in each group has been described in the Results section (P.5, L.176-187).

Point 3: The definition used for mild autonomic dysfunction in T2DM patients (CVRR > 2%) is very wide and could also be applied to the control group. Although the results do show that this group had a significantly lower CVRR than the control group, is there actually a definition for normal CVRR? In general, do all T2DM patients have lower CVRR than normal? In this particular cohort, did any T2DM patients have a CVRR similar or even higher than individuals in the control group? Were there any T2DM patients in this cohort that actually had normal autonomic function? The distinction is very important and I think these should be clarified, otherwise one cannot conclude that the drop in SBP is entirely due to autonomic dysfunction, but rather it would be a difference between T2DM and control, especially since the absolute magnitude of change in both the mild and severe groups were almost identical from Figure 4.

Response 3: As indicated, the CVRR in the mild group was wide (from 2.15% to 4.90%). In the control group, the CVRR ranged from 2.39% to 6.19%, so the CVRR of some patients were higher than those of control subjects. In general, T2DM patients do not necessarily have lower CVRR than healthy subjects. As indicated in reference 16, some T2DM patients had CVRR higher than 4%. In this study, there is no established definition for normal CVRR. This is a limitation of this study. We have described this limitation in the Discussion.However, HR did not increase after meal ingestion in the mild group, otherwise HR increased in the control group. Therefore, it cannot be said that patients in the mild group had normal autonomic dysfunction. Although it is not statistically significant, the bSBP at baseline in patients were a slightly higher than that in the control group. We think that the difference in not only CVRR but also preprandial BP might be related to the difference in the postprandial BP change between T2DM patients and controls. 

P.9, L.326-327. Discussion

There was no established definition of CVRR for non-diabetic subjects.

Point 4: from the graph (Figure 4) it seems there was a larger decrease in cSBP at 60 minutes compared to bSBP, and that cSBP was recovering slightly more than bSBP at 120 minutes (bSBP almost unchanged)... was cSBP@120 different to cSBP@60? Could the authors discuss these observations and what the implications may be? 

Response 4: In the severe group, it appears that there was a larger decrease in cSBP at 60 minutes compared to bSBP. In this study, cSBP was calculated from SBP2 automatically using a digital automated sphygmomanometer. SBP2 was calculated as follows:SBP2 = augmentation index × (SBP – DBP) + DBP.In this study, as described in our Response 6, bDBP was decreased after a meal.We think that cSBP might have been greatly influenced by bDBP. However, there is no statistically significant difference between bSBP and ∆cSBP. In addition, cSBP at 120 minutes was not significantly different to cSBP at 60 minutes. Thus, we cannot present the implications further.We think that additional investigation is needed.

Point 5: Further to the previous point, the current discussion on cSBP results do not really identify the advantage of measuring cSBP as well as bSBP. The one reference currently used to support the need for measuring cSBP (ref 33) did not actually measure central aortic BP.

Response 5: As you pointed out, we have added and described about cSBP in the Discussion as follows:

P.8, L.300-307. Discussion

As mentioned above, in general, cSBP is recognized to be closely related to cardiovascular events compared to peripheral blood pressure, as demonstrated by studies such as ASCOT-CAFE study [12,13]. cSBP is thought as the blood pressure of the aortic root that is directly applied to key organs, such as the heart and brain. Previous reports described that elevation of the cSBP induced coronary arteriosclerosis [33,34]. As the observation and reduction of cSBP contribute to the prevention of cardiovascular events, measurement of not only brachial blood pressure but also cSBP may be useful in the prevention and treatment of cardiovascular diseases [15].

Point 6: although brachial DBP does not pertain to the definition of PPH, it would be interesting to know if it changed after the meal

Response 6: Brachial DBP significantly decreased after meal in all groups. However, there was no significant interaction among the groups, and as you pointed, brachial DBP does not pertain to the definition of PPH. Thus, we did not include the analysis of brachial DBP as a theme in the study.

Point 7: more discussion on the mechanistic side of how autonomic dysfunction can lead to PPH would be helpful

Response 7: As you pointed out, we added and described about the mechanistic side in the Discussion as follows:

P.7, L.233-237. Discussion

That is, in non-diabetic subjects, HR increased significantly after a meal to compensate for the lowering blood pressure, which resulted to a stable postprandial blood pressure. On the contrary, such increase in HR was not observed in diabetic patients. Consequently, this lack of HR increase might lower the blood pressure.

Point 8: a figure legend may be neater than writing the group names above the relevant lines on the figure

Response 8: As you pointed out, we removed the group names from the figure and added the group names in the figure legends as follows:

P.7, L.191-193. and L.196-197.  Figure legends

The control group is indicated by black line and circle (); the mild group by blue line and square (); and the severe group by red line and triangle ().

Point 9: pg 8, line 250, authors stated bSBP at 60 minutes may be due to hypertension rather than degree of autonomic dysfunction. However, CVRR was also independently associated with change in bSBP1hr 

Response 9: We believed that the bSBP at 60 minutes may be due to hypertension rather than a degree of autonomic dysfunction because the standardized coefficients in the multiple regression analysis for predicting ∆bSBP1hr were larger than that of CVRR, as indicated in Table3. We did not sufficiently describe the results of the analysis. Thus, we added related information in the Results section as follows:

P.7, L.213-214. Results

As shown in Table 3, multiple linear regression analyses revealed that bSBP at baseline was a significant predictor of ∆bSBP1hr, which had a greater influence than CVRR.

Point 10: were interactions included in the stepwise regression?

Response 10:No interactions were included in the stepwise regression. The following factors were used as dependent variables: age, body mass index, CVRR, bSBP, and plasma glucose at baseline. We ran collinearity diagnostics for the variables and confirmed that there was no multicollinearity in the models.

Reviewer 3 Report

Abstract

The design of the study is unclear. The authors mention that this is a cross-sectional study, however, they mention that there was a control group indicating a design corresponding to a case-control study or clinical trial. I think what the authors tried to do is rather to analyze the outcome variables according to whether the patients had T2D or not (unclear). Please, clarify.

Introduction

The objective of the study does not mention any control group, but rather describes that a study will be done in patients with T2D.

Material and methods

·         The numbers of study participants in the abstract does not match the numbers provided in the study design and participants section 37 vs 40; 17 vs 20).

·         What was the reason to include healthy controls if the objective of the study was to assess association in type 2 diabetes patients? How ere the “healthy” controls selected and what was the definition of “healthy (free of type 2 diabetes?)? Was an OGTT performed in the healthy control to rule out T2D or IGT?

·         Exclusion criteria are not the opposite of inclusion criteria. First you define who to be included, then you apply the exclusion criteria. Thus, the criteria HbA1c > 12.0% does not make much sense as they are not included in your study population.

·         Healthy controls: Did you apply HbA1C tests for the healthy controls? There may be a risk that they have undiagnosed T2D. Please, add smt about this in the limitations of the study section.

·         Please, make sure to provide a description of each covariate sued in the study. I.e., I did not find any information how BMI was measured/assessed

·         Please, provide information on sample size calculations and how did you end up with the numbers for the control and T2D groups and not the same amount of participants in both groups.

·         Did you run collinearity diagnostics for the variables included in the model?

·         How was completeness and efficiency of the multiple regression models assesses?

·         Did you check whether the continuous variables followed a normal distribution?

Results

·         F values in table 1 are not necessary

·         T values in Table 2 are not necessary

·         Table 3: Please, provide 95% confidence intervals for the slope instead of p-values

·         Please, provide first the unadjusted slopes, then the adjusted values

Author Response

Response to Reviewer 3 Comments

Thank you very much for your review and for your helpful comments. We have revised this manuscript on the basis of your comments and those of the other reviewers.The revised paper has been edited for grammar and journal style by an expert in English medical writing prior to resubmission. Revisions are highlighted in yellow in the manuscript and are underlined belowOur responses to your comments are shown below.

Abstract

Point 1: The design of the study is unclear. The authors mention that this is a cross-sectional study, however, they mention that there was a control group indicating a design corresponding to a case-control study or clinical trial. I think what the authors tried to do is rather to analyze the outcome variables according to whether the patients had T2D or not (unclear). Please, clarify.

Response 1: The purpose of the present study was to investigate differences in the change in postprandial blood pressure in non-diabetic subjects and T2DM patients and patients with mild or severe cardiac autonomic dysfunction. As you pointed out, this was a cross-sectional study, but a control group was included to compare patients. Accordingly, we added related information in the Abstract and Materials and Methods as follows:

P.1, L.17-18.Abstract, and P.2, L.80.Materials and Methods 

This study was conducted as a cross-sectional study with case-control comparison.

Introduction

Point 2: The objective of the study does not mention any control group, but rather describes that a study will be done in patients with T2D.

Response 2: As you pointed out, we have added and described related information in the Abstract and Introduction as follows:

P.1, L.18-20. Abstract

The purpose of this study was to investigate postprandial blood pressure changes in non-diabetic subjects and patients with type 2 diabetes and patients with mild or severe cardiac autonomic dysfunction.

P.2, L.73-77. Introduction

The purpose of the present study was as follows: (1) to investigate differences in the change in postprandial blood pressure in non-diabetic subjectsand patients with T2DM and mild or severe cardiac autonomic dysfunction by measuring brachial systolic blood pressure (bSBP) as well as cSBP and (2) to clarify the relationship between changes in postprandial blood pressure and the severity of autonomic dysfunction. 

Material and methods

Point 3: The numbers of study participants in the abstract does not match the numbers provided in the study design and participants section 37 vs 40; 17 vs 20).

Response 3: We have corrected the number of participants in the Abstract as follows:

P.1, L.20-22. Abstract

Forty patientswith type 2 diabetes mellitus and 20 controlsparticipated in this study. Fifty-twoparticipants underwent a meal tolerance test.

Point 4: What was the reason to include healthy controls if the objective of the study was to assess association in type 2 diabetes patients? How ere the “healthy” controls selected and what was the definition of “healthy (free of type 2 diabetes?)? Was an OGTT performed in the healthy control to rule out T2D or IGT?

Response 4: We have included healthy controls because we wanted to compare non-diabetic subjects and patients with normal and mild autonomic dysfunction and patients with severe autonomic dysfunction. As you pointed out, we have clarified the objective of the study in the Introduction. Moreover, we revised the study objective, which is also indicated in our response 2. In this study, controls were non-diabetic subjects who met the Japan Diabetes Society’s T2DM criteria. Their glucose levels were evaluated and should be free of diabetes, according to the Japan Diabetes Society’s T2DM criteria. As it is inappropriate to label controls as “healthy”, we used “non-diabetic” subjects instead of “healthy” controls. An OGTT was not also performed in this study. This is a limitation of the present study. We have described the limitation in the Discussion as follows:

P.9, L.327. Discussion

Fourth, in the control group, evaluations of OGTT or HbA1c were not performed.

Point 5: Exclusion criteria are not the opposite of inclusion criteria. First you define who to be included, then you apply the exclusion criteria. Thus, the criteria HbA1c > 12.0% does not make much sense as they are not included in your study population.

Response 5: As you pointed out, we removed the criteria HbA1c > 12.0% from the Materials and Methods.

Point 6: Healthy controls: Did you apply HbA1C tests for the healthy controls? There may be a risk that they have undiagnosed T2D. Please, add smt about this in the limitations of the study section.

Response 6: In this study, HbA1c tests for the controls were not performed. Again, we have added the limitation in the Discussion as follows:

P.9, L.327. Discussion

Fourth, in the control group, evaluations of OGTT or HbA1c were not performed.

Point 7:Please, make sure to provide a description of each covariate sued in the study. I.e., I did not find any information how BMI was measured/assessed

Response 7: As suggested, we added and described related information in the methods of measuring BMI and SBP2 in the Materials and Methods as follows:

P.3, L.101-102. Materials and Methods

Before the assessment, height and body weight were measured.

P.4, L.133-134. Materials and Methods

Body mass index (BMI) was calculated using the following equation: BMI = body weight (kg) / height (m)2.

P.3, L.118-121. Materials and Methods

Early systolic pressure (SBP1), late systolic pressure (SBP2), and augmentation index (AI) were also measured and calculated automatically using the following equation by HEM-9000AI: AI = (SBP2 – DBP) / (SBP1 – DBP), SBP2 (mmHg) = AI × (SBP – DBP) + DBP.

Point 8: Please, provide information on sample size calculations and how did you end up with the numbers for the control and T2D groups and not the same amount of participants in both groups.

Response 8: We did not perform power calculations to determine the number of subjects. This is another limitation of the present study, and we added this limitation in the Discussion as follows:

P.9, L.326. Discussion

In addition, we did not perform power calculations to determine the number of subjects.

Point 9: Did you run collinearity diagnostics for the variables included in the model?

Response 9: We ran collinearity diagnostics for the variables and confirmed that there was no multicollinearity in the models.

Point 10: How was completeness and efficiency of the multiple regression models assesses?

Response 10: As you pointed out, we added the model equations in Table 3 legends as follows:

P.6, L.203-205. Table Legends

Models are expressed as follows: ∆bSBP1hr = 32.7 − 0.34 × bSBP + 1.95 × CVRR, ∆bSBP2hrs = 14.9 − 0.20 × bSBP + 2.83 × CVRR, ∆cSBP1hr = 19.8 − 0.22 × cSBP, ∆cSBP2hrs = 8.41 − 0.16 × cSBP +2.98 × CVRR.

Point 11: Did you check whether the continuous variables followed a normal distribution?

Response 11: We have checked that the variables followed a normal distribution.

Point 12: Results

F values in table 1 are not necessary. 

Response 12: As you pointed out, we have removed F values from Table 1.

Point 13: T values in Table 2 are not necessary

Response 13: As you pointed out, we have removed T values from Table 2.

Point 14: Table 3: Please, provide 95% confidence intervals for the slope instead of p-values

Response 14: In line with the comment, we added 95% confidence intervals for the unstandardized coefficients and removed P value inTable 3.

Point 15: Please, provide first the unadjusted slopes, then the adjusted values

Response 15: We are sorry that we cannot provide the unadjusted results, because of the analysis system. We think that this point will be the issue in the future.

Reviewer 4 Report

This was a well written paper describing the results of a study designed to determine if diabetic induced changes in autonomic function affect postprandial blood pressure.  The study was well designed and the methods, results and discussion are for the most part, clear and straight forward.  There are a few places where there needs to be editing and a few details regarding the methods and results that would make the paper clearer to understand.

1)  The sentence at the beginning of the second paragraph on page 2 (introduction) line 60-62 doesn't make sense.  Do the authors means" Further, for blood pressyre measurements, the importance of measuring central systolic blood pressure (cSBP), as we do here, is that this measure is more closely related to cardiovascular events compared to peripheral blood pressure, and is recognized by studies such as the Anglo-Scandanavian Cardiac Outcomes Trial Conduit....."

2)  Were subjects given a specific amount of time to finish their meal so that the time from food intake to the time samples were collected and measurements made was equivalent for all subjects?

3) Please use different symbols for the different conditions in the graphs in figures 3 and 4 so that the reader knows which line is associated with each group.

Author Response

Response to Reviewer 4Comments

Thank you very much for your review and for your helpful comments. We have revised this manuscript on the basis of your comments and those of the other reviewers.The revised paper has been edited for grammar and journal style by an expert in English medical writing prior to resubmission. Revisions are highlighted in yellow in the manuscript and are underlined belowOur responses to your comments are shown below.

Point 1: The sentence at the beginning of the second paragraph on page 2 (introduction) line 60-62 doesn't make sense. Do the authors means" Further, for blood pressyre measurements, the importance of measuring central systolic blood pressure (cSBP), as we do here, is that this measure is more closely related to cardiovascular events compared to peripheral blood pressure, and is recognized by studies such as the Anglo-Scandanavian Cardiac Outcomes Trial Conduit....."

Response 1: Thank you very much for your comments. As you pointed out, the Introduction has been changed accordingly as follows:

P.2, L.62-66. Introduction

Further, for blood pressure measurements, the importance of measuring central systolic blood pressure (cSBP), as performed in this study, is that this measure is moreclosely related to cardiovascular events compared to peripheral blood pressure andis recognized by studies such as the Anglo-Scandinavian Cardiac Outcomes Trial Conduit Artery Function Endpoint (ASCOT-CAFE) study [12,13].

Point 2:Were subjects given a specific amount of time to finish their meal so that the time from food intake to the time samples were collected and measurements made was equivalent for all subjects?

Response 2: In the present study, subjects were those who have completely consumed a meal within 15-20 minutes. Themeasurements after meal ingestion systematically began at 60 minutes from the start of food intake. We think that the start of the measurements were the same for all subjects. As you pointed out, we added related information in the Material and Methods as follows:

P.3, L.107-108.Material and Methods

The participants were those who have completely consumed a meal within 15-20 minutes.

Point 3:Please use different symbols for the different conditions in the graphs in figures 3 and 4 so that the reader knows which line is associated with each group.

Response 3:As you pointed out, we havemodified the Figures 3 and 4. The control group was indicated by black line and circle (●); the mild group by blue line and square (■); and the severe group by red line and triangle (▲).

Round 2

Reviewer 3 Report

Dear authors,

thank you for a resubmitted version addressing some of my points raises.

Please, there is no valid research design as cross-sectional study with case-control comparison. Thus, either conduct a case-control study or a cross-sectional study. In either case, define well the research design.

Your comment that you could not run unadjusted models is not valid as you simple remove the covariates from your models to get the unadjusted slopes of the regression line

In addition, you still not answered how the effectiveness and completeness of the statistical models were assessed and compared. Please, ask for advise from a statistician.

Avoid the term "diabetic or non-diabetic" use patients with diabetes or diabetes patients instead.

Please, explain how the fact that you did not perform OGTT/HbA1C in the controls (in case you decide to do a case-control study) affected the study results (misclassification bias)

How was height or weight measured? Light clothing? Shoes on? Empty pockets

Author Response

Response to Reviewer 3 Comments

Thank you very much for your review and for your helpful comments. We have revised the manuscript on the basis of your comments. The revised paper has been edited for grammar and journal style by an expert in English medical writing prior to resubmission. The revisions in this version of the manuscript are highlighted in green and underlined. The previous revisions are highlighted in yellow. Our responses to your comments are shown below.

Point 1: Please, there is no valid research design as cross-sectional study with case-control comparison. Thus, either conduct a case-control study or a cross-sectional study. In either case, define well the research design.

Response 1: As you pointed out, the design of the present study has been redefined as a case-control study.

Point 2: Your comment that you could not run unadjusted models is not valid as you simple remove the covariates from your models to get the unadjusted slopes of the regression line

Response 2: As suggested, we have added the information related to the unadjusted models in Table 3.

Point 3: In addition, you still not answered how the effectiveness and completeness of the statistical models were assessed and compared. Please, ask for advise from a statistician.

Response 3: We believe that these models were effective because the P values were less than 0.05. However, the coefficients of determination were less than 0.40 in all models: therefore, we believe that their ability to predict ∆BP may not be entirely powerful.

In model 2 for the prediction of ∆bSBP1hr, bSBP and CVRR were independent variables. The standardized coefficient of bSBP was higher than that of CVRR. Therefore, we believe that bSBP influenced ∆bSBP1hr to a larger extent than CVRR.

For prediction of ∆cSBP1hr, cSBP at baseline was the independent variable.

In model 2 for the prediction of ∆bSBP2hrs, bSBP and CVRR were as independent variables. The standardized coefficient of CVRR was higher than that of bSBP meaning that CVRR had a greater effect on the change in blood pressure.

In model 2 for the prediction of ∆cSBP2hrs, cSBP and CVRR were as independent variables. Once again, since the standardized coefficient of CVRR was higher than that of cSBP, we believe that the influence of CVRR on ∆cSBP2hrs was larger than that of cSBP. We have added a more detailed description of these findings to clarify this issue and described related information in the revised Results section. (P.6, L.211-P.7, L.223)

Point 4: Avoid the term "diabetic or non-diabetic" use patients with diabetes or diabetes patients instead.

Response 4: As suggested, we have used "patients with diabetes" and "individuals without diabetes".

Point 5: Please, explain how the fact that you did not perform OGTT/HbA1C in the controls (in case you decide to do a case-control study) affected the study results (misclassification bias)

Response 5: In accordance with your comment, we have added the following discussion of this possible limitation in the Discussion section (P.9, L.334-342):

In the control participants, the plasma glucose before meal ingestion was 99.3±1.81 mg/dl, and none had levels exceeding 110 mg/dl, which is the criterion for impaired fasting glucose (IFG). Because we did not perform an OGTT in the control group, we cannot exclude the possibility that individuals with impaired glucose tolerance (IGT) were included. While this is a limitation, the plasma glucose at 120 minutes after meal ingestion was 108.5±6.59 mg/dl, and no individuals had levels higher than 140 mg/dl. Therefore, we believe that it is unlikely that individuals with IGT were included in the control group and that the probability of misclassification bias is low.

Point 6: How was height or weight measured? Light clothing? Shoes on? Empty pockets

Response 6: We measured the height with the participants’ shoes off, and the weight with light clothing and empty pockets. To address your comment, we have added this information to the revised Materials and Methods section. (P.3, L102-103)

Round 3

Reviewer 3 Report

Dear authors,

thank you for resubmitting your work.

Your study is NOT a case-control study and the description/methodology of how the controls were selected does not correspond to a case-control study. 

The title is "Postprandial Blood Pressure Decrease in Patients with Type 2 Diabetes and Mild or Severe Cardiac Autonomic Dysfunction" and does not mention anything about people without type 2 diabetes

Please, be consistent with the title and the objective of the study. Exclude people without T2D ("control" group) and present your results for the T2D patient only. This would make much more sense.

Statistics: You test the efficiency of a multiple linear regression model by assessing changes in the F-value between different models and the completeness of a model by assessing changes in R-square values. You still have not clarified this in the statistical methods section. 

Author Response

Response to Reviewer 3 Comments

Thank you very much for your review and for your helpful comments. We have revised the manuscript on the basis of your comments. The revised paper has been edited for grammar and journal style by an expert in English medical writing prior to resubmission. The revisions in this version of the manuscript are highlighted in green and underlined. The previous revisions are highlighted in yellow. Our responses to your comments are shown below.

Point 1: Your study is NOT a case-control study and the description/methodology of how the controls were selected does not correspond to a case-control study.

The title is "Postprandial Blood Pressure Decrease in Patients with Type 2 Diabetes and Mild or Severe Cardiac Autonomic Dysfunction" and does not mention anything about people without type 2 diabetes

Please, be consistent with the title and the objective of the study. Exclude people without T2D ("control" group) and present your results for the T2D patient only. This would make much more sense.

Response 1: The design of the present study has been redefined as a cross-sectional study again. We would not like to exclude people without T2D. Therefore, we rephrased the follows as the study design:

This was a cross-sectional study designed to investigate postprandial blood pressure changes in individuals without type 2 diabetes and patients with type 2 diabetes and mild or severe cardiac autonomic dysfunction. (P.1, L.17-20, and P.2, L.80)

Point 2: Statistics: You test the efficiency of a multiple linear regression model by assessing changes in the F-value between different models and the completeness of a model by assessing changes in R-square values. You still have not clarified this in the statistical methods section.

Response 2: In accordance with your comment, we have added related information of statistical methods in the Material and Methods section. (P.4, L153-154)